# Clinical and Ultrasound Efficacy of Topical Hypertonic Cream (Jovita Osmocell^®^) in the Treatment of Cellulite: A Prospective, Monocentric, Double-Blind, Placebo-Controlled Study

**DOI:** 10.3390/medicina60050781

**Published:** 2024-05-08

**Authors:** Antonio Di Guardo, Carmen Solito, Vito Cantisani, Federica Rega, Luca Gargano, Giovanni Rossi, Noah Musolff, Giulia Azzella, Giovanni Paolino, Luigi Losco, Antonia Rivieccio, Elena Campione, Luca Bianchi, Steven Paul Nisticò, Giovanni Pellacani, Carmen Cantisani

**Affiliations:** 1Department of Dermatology, Policlinico Umberto I, Sapienza University of Rome, 00185 Rome, Italy; 2Department of Radiology, Oncology, Anatomo-Pathology, Sapienza University of Rome, 00185 Rome, Italy; 3Unit of Dermatology, IRCCS Ospedale San Raffaele, 20132 Milan, Italy; 4Plastic Surgery Unit, Department of Medicine, Surgery and Dentistry, University of Salerno, 84081 Salerno, Italy; 5Dermatologic Unit, Department of Systems Medicine, University of Rome Tor Vergata, 00133 Rome, Italy

**Keywords:** cellulite, lipodystrophy, ultrasonography, topical treatment, randomized controlled trial

## Abstract

*Background and Objectives:* Cellulite, or edemato-fibro-sclerotic panniculopathy (EFP), is characterized by dermal and hypodermal changes leading to adipose tissue accumulation and compromised venous circulation. This study investigates the efficacy of a hypertonic cream containing concentrated sodium chloride (Jovita Osmocell^®^) in addressing water retention and structural alterations in adipose tissue, aiming to interrupt the cellulite formation process. *Materials and Methods:* A 12-week, prospective, monocentric, double-blind, placebo-controlled study enrolled 30 female subjects with grade II or III cellulite. Patients were randomized to receive hypertonic cream or a placebo. Thigh circumference, ultrasound evaluations, and standardized photographs were collected at baseline, intermediate, and endpoint visits. Adverse events were monitored. *Results:* After 84 days, the hypertonic cream group exhibited a significant reduction in thigh circumference compared to the placebo group (*p* = 0.0037). B-mode ultrasound examinations revealed significant changes in the parameters studied, such as the thickness of the subcutaneous tissue. No statistically significant changes were noticed in the placebo group. Volunteers reported the investigational product’s pleasantness and good anti-cellulite activity, with no reported adverse events. *Conclusions:* The hypertonic cream demonstrated efficacy in reducing thigh circumference, addressing water retention and structural alterations in adipose tissue. The proposed mechanism involves osmosis, releasing accumulated fluids between fat cells, supporting drainage, and reducing inflammation. This study supports the efficacy and safety of hypertonic sodium chloride emulsions in cellulite treatment and confirms safety and user satisfaction.

## 1. Introduction

Cellulite, more accurately termed edemato-fibro-sclerotic panniculopathy (EFP), arises from changes in the dermis and hypodermis [1].

Cellulite is a dermatological issue primarily found in postpubertal females, characterized by alterations in the skin’s surface, particularly in areas where fat is stored more abundantly, such as the thighs, buttocks, and hips [2,3]. Clinically, these changes present as dimples, dents, or nodules, resulting in an uneven texture resembling an “orange peel” appearance, which is typical of cellulite. Various medical terms like gynoid lipodystrophy, nodular liposclerosis, edemato-fibro-sclerotic panniculopathy, adiposis edematosa, dermopanniculosis deformans, and status protrusus cutis are used to describe cellulite, reflecting different views on its underlying causes [4]. Despite being painless, cellulite has significant negative impacts on individuals’ psychosocial well-being due to its aesthetic implications. Females affected by cellulite commonly experience body dissatisfaction, psychological distress, anxiety, and a diminished quality of life [5].

Cellulite remains a multifaceted entity with a complex interplay of anatomical, hormonal, vascular, inflammatory, and lifestyle-related factors [4]. From an anatomical perspective, cellulite is characterized as an architectural disorder within the dermis and subcutaneous tissue, notably prominent in areas such as the gluteal region. Central to this architecture are fibrous septa that tether the dermis to various adipose layers. Imbalances in biomechanical forces among these septa, adipose layers, and the dermis contribute significantly to the formation and appearance of cellulite. Of particular interest is the gender dimorphism observed in fibrous septal orientation and adipose morphology, which predisposes women to a higher likelihood of cellulite manifestation [6]. Estrogen, a key hormone in females, is believed to exacerbate cellulite progression through its influence on adipose tissue and connective tissue remodeling. Furthermore, the vascular hypothesis posits that microvascular alterations and chronic low-grade inflammation play significant roles in cellulite development [7]. This hypothesis suggests that changes in dermal vasculature, akin to those seen in chronic venous stasis, lead to increased capillary permeability and subsequent fluid leakage into interstitial spaces. The resulting tissue edema and hypoxia trigger neovascularization and fibrous septal thickening, ultimately accentuating skin irregularities and contributing to the characteristic appearance of cellulite. The inflammation hypothesis regarding the development of cellulite emerged from the observation of tenderness upon palpation of the affected skin [8]. It has been suggested that mild septal inflammation could contribute to dermal atrophy and that persistent inflammation may be involved in the formation of fibrous septa [9]. In recent work, using light and electron microscopy, it was found that the cellulite-affected area revealed a morphology distinct from typical adipose depots [10]. Sweat glands were observed in association with adipocytes within the cellulite-affected tissues. Specifically, vesicles were identified in the extracellular matrix, suggesting communication between these two distinct components. Proteomic analysis indicated that adipose tissue affected by cellulite exhibits a high level of oxidative stress and undergoes remodeling phenomena [10]. In addition to anatomical and vascular considerations, lifestyle factors such as diet and physical activity significantly influence cellulite severity [11]. High-carbohydrate diets and sedentary lifestyles may exacerbate cellulite development by promoting hyperinsulinemia, stimulating lipogenesis, and increasing overall body fat content. Moreover, aging exacerbates cellulite through dermal atrophy and hypertrophy of fat lobules, further complicating its management in elderly individuals, particularly those with a high body mass index (BMI). Understanding the complex nature of cellulite is crucial for developing effective management strategies. By addressing these diverse mechanisms underlying cellulite pathogenesis, clinicians can offer more personalized and effective interventions to alleviate the burden of this common cosmetic concern.

The aim of this study is to evaluate the safety and efficacy of a topical treatment primarily targeting the accumulation of fluids in the interstitial space caused by microcirculatory alterations. Under normal conditions, fat cells (adipocytes) in the subcutaneous tissue serve as an energy reserve for the body, utilized when needed. When venous circulation is compromised, this reserve becomes less accessible, leading to the accumulation of adipose tissue [6,12]. This accumulation compresses blood capillaries, causing them to leak plasma through porous walls. Over time, this plasma causes inflammation and fibrosis in subcutaneous tissues, further compressing capillaries. As their walls become more porous, fluid leaks out into the interstitial space between fat cells [13,14]. With time, the drainage of excess fluid becomes increasingly challenging, creating a self-sustaining “vicious cycle”. The osmolarity of the trapped interstitial fluid is similar to that of plasma (mOsm/L = 281 vs. 280). A solution with 0.9% sodium chloride, often called isotonic, has a slightly higher osmolarity than blood (around 300 mOsm/L), making it slightly hypertonic. Consequently, the application of solutions with sodium chloride concentrations greater than 0.9% (hypertonic) may activate an osmotic process, facilitating the drainage of excess fluids trapped between fat cells, addressing the primum movens of the condition [15]. The exclusivity of the formulation of Jovita Osmocell^®^ is based on an emulsion able to stably incorporate a high concentration of sodium chloride (13%), which makes it highly hypertonic. The aim of this medical device is to contrast the primary factors contributing to cellulite, namely water retention and structural alterations in adipose tissue. The intervention begins with a hypertonic emulsion designed to remain on the skin’s surface. This emulsion mechanically displaces excess interstitial fluids, facilitating the restoration of physiological conditions necessary to interrupt and reverse the process leading to cellulite formation and exacerbation. The device comprises a stable, highly hypertonic emulsion containing a concentrated amount of sodium chloride. Upon contact with the skin (a semipermeable membrane), the emulsion employs osmosis to release accumulated fluids between fat cells, directing them outward. This process supports subsequent elimination and, consequently, reduces inflammation in the adipose tissue. The primary aim of the study was to validate the efficacy of the topical product containing hyper concentrated sodium chloride in female patients with grade II or III cellulite in the thighs and buttocks, according to the Nürnberger–Müller classification [16]. The measurement of thigh diameter at the end of the 12-week treatment served as the primary outcome. The evaluation of epidermal and dermis thickness, as well as the observation of the morphology of the upper layer of the subcutaneous tissue and dermis, represent the ultrasound endpoint. Additionally, thigh diameter at intermediate follow-up and any adverse events related to product usage were also assessed.

## 2. Materials and Methods

We performed a prospective single-center, blinded, two-arm controlled study to evaluate the clinical efficacy, the risk–benefit ratio, and the presence of any adverse reactions of hypertonic cream treatment in comparison with a placebo cream (Jovita Osmocell^®^, Laboratori Farmaceutici Krimy Spa, Monterotondo, Rome, Italy). Consent was obtained immediately after the participants were provided with information about the objectives of the study and the methods put in place to achieve them by the investigators through a specific and detailed Information Sheet. The experimental study and its documentation were submitted to and subsequently approved by the relevant Ethics Committee. The study was conducted in accordance with GCP, ethical principles derived from the Declaration of Helsinki, and current regulations on observational studies. The protocol was approved by the Ethics Committee of Sapienza University of Rome, Policlinico Umberto I Hospital (approval code: prot. 0582/2022; approval date: 20 July 2022). The criteria for inclusion in the study were female subjects; age > 18 years; presence of grade II or III cellulitis according to the Nürnberger–Müller scale; not needing or not scheduled for other treatment and therefore scheduled for clinical follow-up according to current guidelines; immunocompetent. Exclusion criteria were subjects who have undergone other treatment; subjects with diabetes mellitus undergoing insulin treatment; previous diagnoses of immunologic and/or autoimmune skin diseases (SLE, scleroderma, etc.); other skin diseases in the area to be treated that could affect evaluations or outcomes; subjects unable to provide consent or not deemed fit to follow the directions given; any contraindications/allergies to elements/excipients contained in the study products. Thirty patients (mean age 50.6 years) were enrolled after the investigator checked all inclusion and non-inclusion (T0) criteria. Patients were randomized into the treatment arm (hypertonic topical) and placebo arm (vehicle cream without active component) in a 1:1 ratio and received the product to be applied and information on the mode of application. At the baseline visit, the following characteristics of the subject were recorded: age, sex, phototype (according to Fitzpatrick), history of other relevant skin or general diseases not excluded from the study, chronic drug use, smoking and alcohol habit/history, and body weight. At the baseline and follow-up visits, the following information was acquired: diameter measurement of the sample thigh selected for the study (the circumference measurement was performed on each posterior thigh, at 10 cm below the gluteus fold; a flexible but inelastic anthropometric tape was used to take the circumference measurement); a standardized photographic image using V-Track (Vidix^®^, Canfield Scientific Italia, Spilamberto, Italy); body weight; ultrasound evaluation of the thickness and echogenicity of the subcutaneous adipose tissue performed by means of baseline (B) mode at the level of the trochanteric region; color-Doppler US examination of the lower limbs, with a linear probe at different frequency (7–14 MHz) was subsequently performed.

Classical ultrasound examinations were performed using a Mindray ultrasound (Shenzhen Mindray Bio-Medical Electronics Co., Ltd., Shenzhen, China). The study used a 7–14 MHz linear probe.

The standard ultrasound examination in B mode, performed by qualified staff with at least twenty years of experience, was performed three times in all women: before the start of anti-cellulite therapy (at time T0), after 42 days of treatment (at time T1) and after its completion (84 days; at time T2). The examinations were carried out on the trochanteric region of the thigh, always in the same position, considering the ultrasound reference bone markers, both in axial and longitudinal scanning, taking the presence of the great trochanter, the bone echo of the femoral neck, and the proximal part of the femoral diaphysis as reference.

In B mode, the following parameters were evaluated: epidermal thickness, dermis thickness, total thickness, and the echogenicity of subcutaneous tissue. Most of the parameters, that is, the thickness of individual structures, were measured by designation on the ultrasound image line between the beginning and the end of each structure. Parameters such as echogenicity and the presence or absence of edema were evaluated with eye-size as more hypoechoic. The integration with the Color-Doppler US examination of the lower limbs allowed the venous vessels of the legs to be visualized and studied in order to detect the presence of any venous pathologies affecting the lower limbs. The patients underwent two follow-up visits every 6 weeks (T1 42 days ± 10; and T2 84 days ± 15). Any adverse events related to the use of the device were recorded at each follow-up visit. The two products (hypertonic cream and vehicle cream) were supplied as creams of similar appearance and texture. All the samples were blindly assigned a code before being stored at ambient humidity and temperature in their original container. Hypertonic cream and the placebo were applied once daily on the right or left site in a double-blind fashion for 84 days.

## 3. Results

The average age of the participants was 50.6 years. All individuals were of Caucasian ethnicity, with 37% being classified as Fitzpatrick skin type II and 63% as type III. Eighteen patients had a body mass index (BMI) of 25 or higher (ten in the treated group, eight in the placebo group). Thirty percent of the participants were smokers (six in the hypertonic cream group and three in the placebo group), while 10% reported occasional alcohol consumption (two in the hypertonic cream group and one in the placebo group). The main epidemiological characteristics are outlined in Table 1.

Of the 30 subjects enrolled, there was a 10% loss in the sample. In the arm treated with hypertonic cream, thigh circumference was significantly reduced (at T1 *p* = 0.027, paired *t*-test; at T2 *p* = 0.0037, paired *t*-test). The mean reduction in patients treated with the medical device was −1.08 cm and −2.12 cm after 42 and 84 days of treatment, respectively. Placebo application reduced thigh circumference by −0.02 cm (day 84) (Figure 1). Therefore, the results of the placebo arm were not statistically significant compared with the baseline value (*p* = 0.43; paired *t*-test). At the end of 84 days of treatment, the thigh circumference in the treated group was 3.35 cm lower than that in the placebo group (61.10 cm vs. 64.45 cm) (Figure 1). A lateral view of the left thigh of a patient treated with hypertonic cream is shown in Figure 2.

In other cases, there was mainly an improvement in skin tone (Figure 3).

Additionally, the efficacy of hypertonic topical treatment was assessed, specifically in patients with a BMI greater than or equal to 25. A notable significant reduction in thigh diameter was observed within this subgroup of patients (*p* = 0.0013; paired *t*-test; mean circumference reduction: 1.74 cm and −2.97 cm after 42 and 84 days of treatment, respectively).

B-mode ultrasound examinations revealed significant changes in the parameters studied. The thickness of the subcutaneous tissue was reduced, confirmed by the results obtained from measuring the circumference of the thigh (Figure 4). The average reduction in the thickness of the subcutaneous tissue in the group using cellulite cream was 2 mm after 42 days and 3 mm after 84 days of treatment. No statistically significant changes were noticed in the placebo group (Figure 5). The analysis of the thickness of the dermis showed that, following the anti-cellulite treatment used, no substantial changes were found. Similarly to the case of the thickness of the subcutaneous tissue, no significant differences were noted in women taking the placebo. As for the thickness of the epidermis after the completion of treatment, no differences were observed in both the hypertonic cream group and the placebo group before and after the completion of treatment. The comparison of echogenicity before and after anti-cellulite treatment showed an improvement in subcutaneous tissue echogenicity in 25% of the patients with hypertonic cream according to the index of reduction of edema of subcutaneous tissue. In the placebo group, the echogenicity did not change. In addition, the analysis of vascular-connective shoots showed that in cases of moderate–severe cellulite (grade II or III according to the Nürnberger–Müller scale), the latter showed an irregular and discontinuous trend, and after the application of the device, there was an improvement in the regularity of the same in 20% of the hypertonic cream group. No significant differences were noted in women taking the placebo. The additional color-doppler US examination showed that 53% of patients showed abnormalities in the venous system of the lower limbs, particularly regarding the presence of perforating veins such as Cockett B in 25% of patients, Boyd in 15% of patients, and Huxley in 1% of patients.

The opinion of volunteers confirmed the pleasantness and the good anti-cellulite activity of the investigational product. No adverse events were reported.

## 4. Discussion

The present study aimed to evaluate the efficacy of a hypertonic cream containing a concentrated amount of sodium chloride as a topical treatment for female patients with grade II or III cellulite in the thighs and buttocks. Edemato-fibro-sclerotic panniculopathy involves changes in the dermis and hypodermis, leading to the accumulation of adipose tissue and compromised venous circulation.

Cellulite is a common cosmetic concern, and over time, various treatment options have been proposed [4]. These treatments target different aspects of cellulite’s development and progression, including impaired microcirculation, drainage deficiencies, architectural disturbances, and skin laxity. Among the available treatments are topical agents; oral supplements; massages; energy-based therapies such as radiofrequency (RF), laser, and light therapy; acoustic wave therapy; subcision; and injectable treatments like dermal fillers and biologics. Topical agents containing ingredients like caffeine [17] and retinol [18] are commonly used to improve cellulite appearance by stimulating microcirculation, promoting lipolysis, and increasing dermal thickness. Placebo-controlled, randomized studies have shown notable enhancements in cellulite severity with topical preparations containing caffeine and/or retinol [18]. Herbal Emgel, a compress that encapsulates 100–200 g of mixed herbs, was shown to be effective in reducing the severity of cellulite in 21 premenopausal women [19]. However, these studies were limited in size and duration. A systematic review and meta-analysis evaluating the effectiveness of topical products for reducing cellulite found moderate efficacy in reducing thigh circumference [20]. However, despite some promising results in small studies, these topical agents lack robust evidence for long-term efficacy and are not FDA-approved for cellulite treatment. Oral supplements containing antioxidant-rich ingredients like Vitis vinifera and Ginkgo biloba are believed to improve skin appearance, but clinical evidence supporting their effectiveness in treating cellulite is limited [21]. The potential of *Rosmarinus officinalis* in the treatment of cellulite has been investigated due to its potential anti-inflammatory properties [22]. Similarly, massage, both manual and mechanical, aims to stimulate lymphatic drainage and improve microcirculation but is not commonly used in clinical practice [23]. Energy-based therapies like RF, laser, and light therapy work by delivering thermal energy to the target area, promoting collagen remodeling, and tightening the skin [6,24]. While some devices have shown efficacy in reducing cellulite appearance, multiple treatment sessions are usually required [4]. Acoustic wave therapy, including radial and focused shock waves, aims to improve microcirculation and stimulate collagen production [25]. While several studies have reported improvements in cellulite appearance with these therapies, the durability of results beyond one year is uncertain. Microwave therapy (frequency range between 1 and 300 GHz) on the buttocks and back of the thighs has been shown to be effective in improving the degree of cellulite [26]. Microwave therapy acts on the solubilization of deeper collagen fibers, stimulates fibroblast activation, and induces remodeling of collagen fibers through controlled hyperthermia. Subcision, a surgical technique that severs fibrous septal bands tethering the skin to underlying tissue, can effectively treat cellulite depressions [9]. Manual, vacuum-assisted, and laser-assisted subcision techniques have shown promising results, but adverse effects like bruising and pain are common [4]. Injectable treatments, including collagenase injections and dermal fillers like calcium hydroxyapatite and poly-l-lactic acid, offer additional options for cellulite treatment [27,28]. These treatments aim to stimulate collagen production and improve skin texture and laxity, but their long-term efficacy for cellulite remains to be fully elucidated.

In this study, on the other hand, we focused on addressing the primary factors contributing to cellulite formation, especially water retention and structural alterations in adipose tissue. The mechanism of action proposed for the hypertonic cream involves the use of osmosis to release accumulated fluids between fat cells, facilitating drainage and reducing inflammation in adipose tissue. Solutions containing concentrations of sodium chloride higher than 0.9% (hypertonic) are able to activate an osmotic process; seawater’s osmotic activity containing a high concentration of salts equal to 3.5% has been demonstrated [15]. However, despite such a high salt concentration, seawater’s osmotic function proves to be very light, as it is strongly compromised by the known inability of salts to permeate the stratum corneum. The high affinity of the salt (active substance) with the vehicle (water) reveals its tendency to remain in the excipient and not to migrate to the skin, resulting in a low value of the partition coefficient of all solutions. In addition, the skin is totally refractory to water, which in no way can interfere with the degree of permeability of the barrier. In contrast to the solutions in the Jovita Osmocell^®^ emulsion, the use of lipophilic substances significantly increases the value of the partition coefficient by reducing the binding energy between the active substance and the vehicle. This means that unlike aqueous solutions, in the emulsion, the sodium chloride has a reduced affinity with respect to the vehicle and therefore a higher rate of release on the skin. The high concentration of sodium chloride, 13%, present in Jovita Osmocell^®^ and a correct coefficient of partition between the lipophilic and hydrophilic phase allows the hypertonic emulsion to perform a powerful mechanical anti-cellulite action by osmosis.

A 12-week treatment period was selected to assess the long-term effects of the hypertonic cream on thigh circumference, with measurements taken at baseline, intermediate follow-up (42 days), and the study endpoint (84 days). The results indicate a statistically significant reduction in thigh circumference in the group treated with the hypertonic cream compared to the placebo group. The observed reduction in thigh circumference of 3.35 cm in the treated group compared to the control group at the end of 84 days is noteworthy. Moreover, within the subgroup with BMI greater or equal to 25, a greater reduction in thigh diameter than in normal-weight women was observed.

Ultrasound is a widely used diagnostic method in the medical field. In recent years, more and more attempts have been made to apply it in dermatology and aesthetic medicine, which is a rapidly developing field [29,30,31]. The results of the research presented above explicitly indicate that both classic and high-frequency ultrasonography can be applied in the examination of the skin. Among the parameters determined by means of B-mode US, the most relevant was subcutaneous tissue thickness, which decreased after the anti-cellulite treatment. Microcirculation improvement was probably responsible for reducing subcutaneous tissue thickness. All parameters evaluated using high-frequency ultrasonography, except for the epidermal thickness, may be regarded as useful in anti-cellulite treatment monitoring. Measuring epidermal thickness creates a possibility of miscalculation because the layer is very thin. Additionally, the ultrasound image of the epidermis consists of the echo created between the ultrasound gel and the skin surface as well as the epidermal echo [32]. A hyperechoic fascicle visible on the ultrasound image is not the epidermis ‘sensu stricto’; therefore, an objective measurement in this case is impossible [32]. According to our present study, the most useful parameters of cellulite evaluation are the thickness, echogenicity, and the presence of the fibro-connective shoots of the subcutaneous tissue; the echogenicity and thickness of the dermis. The measurement of the thickness of the subcutaneous tissue before and after treatment explicitly indicates that the presence and irregularity of fibro-connective beams indicates a more advanced degree of the condition of cellulite.

As is known, in fact, in pannicolopathy, we initially find a condition of edema, generally distributed at the level of the most superficial subcutaneous adipose tissue, which first shows the pathological signs of pannicolopathy, which at ultrasound has a more hypoechoic appearance. The connective branches of the subcutaneous tissue, at the initial stage, are still linear, but they can, as soon as the condition of fibrosis begins, thicken, and therefore become thicker and more visible. In fibrosis, in fact, an increase in the thickness of subcutaneous adipose tissue occurs and the loose cell tissue begins to increase its echogenicity, in relation to the condition of fibrosis from reduced blood perfusion; the connective branches tend to assume a wavy appearance and continue to increase in thickness, until they become discontinuous, as is observed in the conditions of sclerosis. The final stage of panniculopathy is represented by the condition of fibrosclerosis, that is, a compaction of the fibrous mass at the level of the subcutaneous adipose tissue, which in the ultrasound examination takes on a more hyperechogenic appearance, as it is increasingly rich in collagen. The connective branches, at this stage, will be almost indistinguishable from the fibrosis condition of the tissue that surrounds them, with the appearance of sclerotic nodules. In sclerosis, however, there is a complete subversion of the echogenicity of adipose tissue towards a hepatitis picture, then a homogeneously hyperechoic picture, where a cleavage plan with the underlying muscle is poorly appreciable. The improvement in echogenicity, understood as the reduction in the condition of edema, in the earlier states of the pannicolopathy, together with the improvement in the course and distribution of the fibro-connective branches of subcutaneous adipose tissue, which are more straight and less diastased, observed after therapy, is a sign of a reduction in the condition of panniculopathy [33]. However, in our study, the only sign which was observed in treated patients was the reduction in subcutaneous thickness. Meanwhile, with regard to other possible signs, we did not observe a direct correlation with the clinical appearance on the one hand, and on the other, we still necessitate the use of computerized objective software which permit the quantification of echogenicity. Therefore, further studies are necessary to create examination standards.

The application of hypertonic cream may effectively target water retention and structural alterations in adipose tissue, supporting the hypothesis that hypertonic solutions can activate an osmotic process, facilitating the drainage of excess fluids. These effects are greater than those obtained with other anti-cellulite treatments like radiofrequency energy, infrared light, and mechanical manipulation [6]. The absence of reported adverse events and the confirmation of the pleasantness and anti-cellulite activity of the investigational product by volunteers are positive indicators of the safety and user satisfaction associated with the hypertonic cream.

Certain limitations must be considered when interpreting our findings. Firstly, our comparison of the effectiveness of hypertonic cream with a placebo cream was exploratory in nature, aimed at assessing the efficacy of the product. Future comparative explanatory studies are warranted to evaluate the efficacy in comparison with anti-cellulite standard topical products. Secondly, we did not document the menstrual cycle phase of the subjects during study visits. It is recognized that cellulite can be affected and exacerbated by cyclic edema. Nevertheless, given that our study involved intra-patient evaluation, this potential confounding factor could have influenced the effect of both study products similarly. Further studies are needed to evaluate the long-term maintenance of the outcome. In addition, other factors such as cigarette smoking, alcohol status, or medication intake could also be a limitation in data analysis. It is essential to acknowledge the relatively small size of the sample and the 10% loss in the study population.

## 5. Conclusions

Our study demonstrated the efficacy and safety of using hypertonic sodium chloride emulsions in the treatment of cellulite. Future research with a larger and more diverse sample may provide additional insights into the generalizability of these findings. Moreover, the study primarily focused on objective measurements such as thigh circumference, ultrasound evaluations, and standardized photographic images. While these quantitative assessments are valuable, incorporating subjective measures, such as patient-reported outcomes or quality-of-life assessments, could enhance the comprehensive evaluation of the hypertonic cream’s effectiveness from both a clinical and patient perspective.

## Figures and Tables

**Figure 1 medicina-60-00781-f001:**
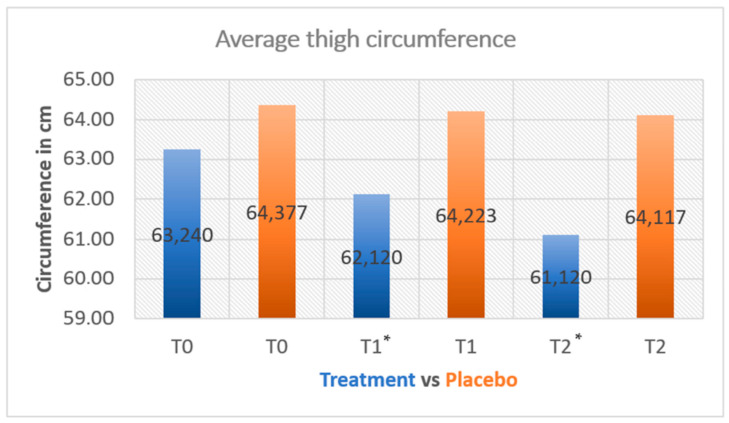
Average circumference in centimeters of patients treated with hypertonic cream vs. placebo. * statistically significant (*p* < 0.05).

**Figure 2 medicina-60-00781-f002:**
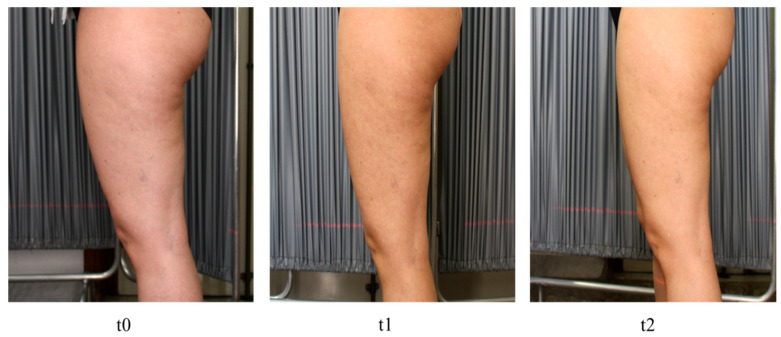
Lateral view of the left thigh of a patient treated with hypertonic cream at 0, t1, and t2.

**Figure 3 medicina-60-00781-f003:**
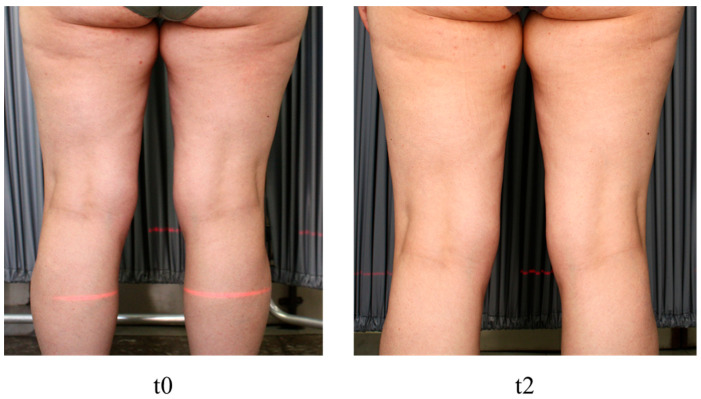
Posterior view of the thighs of a patient treated with hypertonic cream. Improvement in skin tone and reduction in dimples between t0 and t2 can be observed.

**Figure 4 medicina-60-00781-f004:**
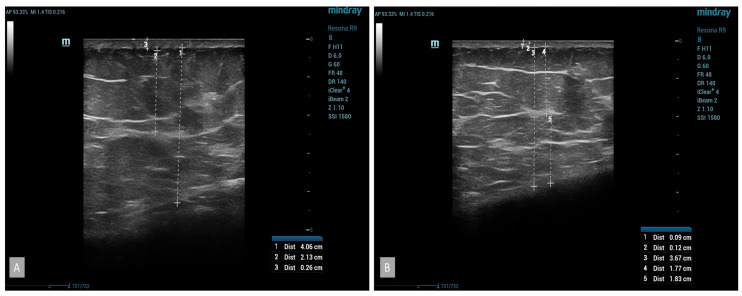
At baseline US, subcutaneous thickness with fibro-tissues bands was evident (**A**); after treatment, reduction in subcutaneous thickness and an increase in the echogenicity of the hypodermis were evident (**B**).

**Figure 5 medicina-60-00781-f005:**
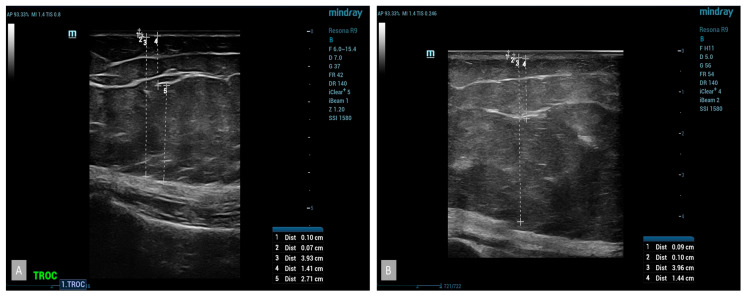
Placebo, at baseline US, pre (**A**) and post treatment (**B**) evaluation of placebo patient did not show any significant changes.

**Table 1 medicina-60-00781-t001:** Epidemiological characteristics of enrolled patients. BMI: body mass index.

Characteristic	All Patients *n* (%)	Patients Treated with Jovita Osmocell^®^ *n* (%)	Patients Treated with Placebo Cream *n* (%)
Number of patients	30	15	15
Sex			
Females	30 (100%)	15	15
Males	0	0	
Age (mean) in years	50.6	52.9	49.4
Smoking status			
Yes	9 (30%)	5	4
No	21 (70%)	10	11
Caucasian	30 (100%)	15	15
Fitzpatrick II	11 (37%)	4	7
Fitzpatrick III	19 (63%)	11	8
BMI (kg/m^2^)			
Underweight (≤18.4)	0	0	0
Normal range (18.5–24.9)	12 (40%)	5	7
Overweight–pre-obese (25.0–29.9)	12 (40%)	6	6
Obese (Class I) (30.0–34.9)	5 (16.67%)	3	2
Obese (Class II) (35.0–39.9)	1 (3.33%)	1	0
Obese (Class III) (≥40.0)	0	0	0
Smoking status			
Yes	9 (30%)	6	3
No	21 (70%)	9	12
Alcohol consumption			
Regular	0	0	0
Occasional	3 (10%)	2	1
No alcohol consumption	27 (90%)	13	14

## Data Availability

All data reported in the present manuscript will be available on request from the authors.

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
