# Peer review of "Clinical and Ultrasound Efficacy of Topical Hypertonic Cream (Jovita Osmocell®) in the Treatment of Cellulite: A Prospective, Monocentric, Double-Blind, Placebo-Controlled Study"

_medicina, 2024, doi:10.3390/medicina60050781_

Round 1
Reviewer 1 Report
Comments and Suggestions for Authors
Paper can be interesting but it has serious flaws.
1. Specify the type of study due to the strobe checklist and add as a supplementary material
2. literature is very poor please extend the reference section
3. Introduction , Discussion is too short please extend
4. it is hard to Derive such conclussions , Please mention something about limitation which are huge in this study
english is fine some minor spell check requiered
please add more photogrqphs and graphical content in main document and in supplementary as well
thank you
Comments on the Quality of English LanguageEnglish is fine some minor corrections are necessary for better quality of this paper
Author Response
Dear Reviewer,
We appreciate your attention and invaluable suggestions. In compliance with your recommendations:
- We have uploaded the strobe checklist to provide additional information on the study type.
- The literature has been significantly expanded in accordance with your suggestion.
- The "Introduction" and "Discussion" sections have been significantly expanded and deepened.
- The main limitations of the study have been outlined in the final part of the "Discussion" section.
Furthermore, a spell check has been conducted to refine the English.
As requested by you, additional figures have been added to the main manuscript.
Sincerely
Reviewer 2 Report
Comments and Suggestions for Authors
This article is very interesting but it contains many problems (see below).
Major points
1) Materials & Methods: Please compare strictly the two groups (hypertonic cream group and placebo group). Table shows the clinical characteristics of 30 enrolled patients. Please describe the two groups separately. We cannot understand of these of two groups separately.
2) Materials & Methods: Please describe clearly and in detail how to perform ultrasound examination. Determination of scanning planes, and experience of examiners. No intra-and inter-observer variability?
3) Discussion: Detailed interpretation of ultrasound images is usually very laborious. Please explain in detail what tissue conditions "echogenicity" and "presence of the vascular-connective shoots" refer to.
4) Discussion: Please explain how to evaluate accurately “echogenicity”.
5) Discussion: I cannot understand what means “presence and irregularity of the vascular beams-connective tissue indicates a more advanced degree of the condition of cellulite”. Please explain in detail.
6) Figures: Please show representative ultrasound of the thickness, echogenicity, and the presence of the vascular-connective shoots of the subcutaneous tissue, echogenicity and thickness of the dermis.
7) Figures: Please add some representative ultrasound images before and after treatment.
Minor points
1) The terminology: Why “control group” and “placebo group” ? Two-arm controlled study means “hypertonic cream group” and “placebo group”. What is “control group”? It’s confusing to the reader.
2) English: To be revised. Many spelling errors. Eco-color-doppler? Ecocolor-doppler? I also think that "doppler" should be written as "Doppler".
3) Results: Please show “M+/-SD”, not “M” only.
Comments on the Quality of English LanguageEnglish: To be revised. Many spelling errors. Eco-color-doppler? Ecocolor-doppler? I also think that "doppler" should be written as "Doppler".
Author Response
Asnwer to the editor
Dear Reviewer,
Thank you for your attention and invaluable suggestions. In compliance with your recommendations:
Major points
1) Materials & Methods: Please compare strictly the two groups (hypertonic cream group and placebo group). Table shows the clinical characteristics of 30 enrolled patients. Please describe the two groups separately. We cannot understand of these of two groups separately.
2) Materials & Methods: Please describe clearly and in detail how to perform ultrasound examination. Determination of scanning planes, and experience of examiners. No intra-and inter-observer variability?
See the corrections made in the manuscript
No interoberserv variability was assessed
3) Discussion: Detailed interpretation of ultrasound images is usually very laborious. Please explain in detail what tissue conditions "echogenicity" and "presence of the vascular-connective shoots" refer to.
The echogenicity was compared to dermis, and when edems is presenti t appears more hypoechoic while when the treatment is applied it appears mode hypercogenic. With the presence of "presence of the vascular-connective shoots" we refere to what is better to call fibrous-connective shoots or bands and we apologize and corrected it within the text.
4) Discussion: Please explain how to evaluate accurately “echogenicity”. We did evaluate eye-size in fact the main and objective used finding was the thickness. In the future we hope to have softwares with estimate objectively echogenicity
5) Discussion: I cannot understand what means “presence and irregularity of the vascular beams-connective tissue indicates a more advanced degree of the condition of cellulite”. Please explain in detail.
As we already explained it was a typos error, we usually refere to fibro-connective tissues bands and when they are thickened, several and irregular it correpond to worse cellulite
6) Figures: Please show representative ultrasound of the thickness, echogenicity, and the presence of the vascular-connective shoots of the subcutaneous tissue, echogenicity and thickness of the dermis.
We did include as requested
7) Figures: Please add some representative ultrasound images before and after treatment.
Thanks for the request and we did include those images
Minor points
1) The terminology: Why “control group” and “placebo group” ? Two-arm controlled study means “hypertonic cream group” and “placebo group”. What is “control group”? It’s confusing to the reader.
2) English: To be revised. Many spelling errors. Eco-color-doppler? Ecocolor-doppler? I also think that "doppler" should be written as "Doppler".
3) Results: Please show “M+/-SD”, not “M” only.
Comments on the Quality of English Language
English: To be revised. Many spelling errors. Eco-color-doppler? Ecocolor-doppler? I also think that "doppler" should be written as "Doppler".
Sincerely
Round 2
Reviewer 1 Report
Comments and Suggestions for Authors
Accept as it is
Author Response
Dear Reviewer,
It has been a pleasure collaborating with you. We extend our heartfelt gratitude for your invaluable guidance and dedication of time.
Sincerely,
Carmen Cantisani
Reviewer 2 Report
Comments and Suggestions for Authors
This article has been fully revised, and it is acceptable for publication in the present form. The reply was also carefully written and understood.
1) Sonographically there are some strange expressions (eco-color-Doppler, classic ultrasonography).
2) Fig.5.left and right. The magnification is not the same.
Sincerely yours,
Hideaki Ishida
Comments on the Quality of English Language
-
Author Response
Dear Reviewer, We modified the manuscript as requested, unforming the term B-mode US and Color-Doppler-US instead of Eco-cholor-doppler or classic ultrasonography; you can find in the manuscript as underlined in yellow. Regarding the fig.5 observations, although the magnification is different the results are clearly visible and therefore we would like to mantain this image that was indicative to clearly show the difference in measurement and all the US features. Many thanks. I send to you my best regards. Yours, Vito Cantisani